# PDE10A Inhibition Reduces NLRP3 Activation and Pyroptosis in Sepsis and Nerve Injury

**DOI:** 10.3390/ijms26104498

**Published:** 2025-05-08

**Authors:** Bradford C. Berk, Camila Lage Chávez, Chia George Hsu

**Affiliations:** 1Department of Physical Medicine and Rehabilitation, University of Rochester School of Medicine and Dentistry, Rochester, NY 14642, USA; 2Department of Medicine, University of Rochester School of Medicine and Dentistry, Rochester, NY 14624, USA; camila_lage@urmc.rochester.edu; 3Department of Kinesiology, The University of Texas at San Antonio, San Antonio, TX 78249, USA; 4Department of Molecular Microbiology and Immunology, The University of Texas at San Antonio, San Antonio, TX 78249, USA

**Keywords:** phosphodiesterase 10A, NLRP3 inflammasome, macrophage, sepsis, nerve injury

## Abstract

Cell death and inflammation are key innate immune responses, but excessive activation can cause tissue damage. The NLRP3 inflammasome is a promising target for reducing inflammation and promoting recovery. Immunometabolism regulates NLRP3 responses in neurological and inflammatory diseases through cyclic nucleotide signaling. Targeting phosphodiesterases (PDEs), which hydrolyze cAMP and cGMP, offer a novel approach to mitigate inflammation. While 14 PDE inhibitors are FDA-approved, PDE10A’s role in NLRP3 inflammasome activation remains unclear. This study investigates the effects of PDE10A inhibition on inflammasome-driven inflammation using two PDE10A inhibitors, MP-10 and TP-10, in macrophage and animal models of sepsis and traumatic nerve injury. Our results show that PDE10A inhibition reduces inflammasome activation by preventing ASC speck formation and by lowering levels of cleaved caspase-1, gasdermin D, and IL-1β, which are key mediators of pyroptosis. In the sepsis model, MP-10 significantly reduced inflammation, decreased plasma IL-1β, alleviated thrombocytopenia, and improved organ damage markers. In the nerve injury model, PDE10A inhibition enhanced motor function recovery and reduced muscle atrophy-related gene expression. These findings suggest that PDE10A inhibition could be a promising therapeutic approach for inflammatory and neuromuscular injuries. Given MP-10’s established safety in human trials, Phase 2 clinical studies for sepsis and nerve injury are highly promising.

## 1. Introduction

Inflammation is a critical response to injury and infection, but when it becomes unregulated, it can lead to tissue damage and contribute to the development of a wide range of diseases, including neuromuscular disorders, cardiovascular diseases, and immune dysfunction [1,2,3,4]. Central to the inflammatory process is the activation of the inflammasome, a multiprotein complex that triggers caspase-1 activation, leading to the production of pro-inflammatory cytokines such as IL-1β and IL-18 [5,6]. This process not only amplifies the inflammatory response but also induces a form of programmed cell death known as pyroptosis, which further exacerbates tissue damage and promotes disease progression [7,8,9]. Consequently, targeting inflammasome activation has emerged as a promising therapeutic strategy to mitigate excessive inflammation and improve outcomes in inflammatory diseases. However, while general anti-inflammatory therapies aim to reduce inflammation systemically, they often lack specificity, resulting in limited efficacy and side effects [6].

Phosphodiesterases (PDEs) are enzymes that regulate intracellular levels of cyclic nucleotides, such as cyclic AMP (cAMP) and cyclic GMP (cGMP), by hydrolyzing them to their inactive forms [10]. These molecules play essential roles in various cellular processes, including immune cell activation and the regulation of inflammation. Given the diversity of PDE isoforms and their involvement in a wide range of physiological functions, PDEs represent promising pharmacological targets [11]. By 2024, a number of PDE inhibitors, including those targeting PDE4 and PDE5, have been FDA-approved for the treatment of various conditions [12]. For instance, sildenafil (Viagra), a PDE5 inhibitor, is used to treat erectile dysfunction by inducing smooth muscle relaxation and increasing blood flow [13]. Similarly, roflumilast, a PDE4 inhibitor, has been approved for managing exacerbations in chronic obstructive pulmonary disease (COPD) [14]. PDE10A, which has a unique expression profile in immune cells, regulates both cAMP and cGMP levels, and is emerging as a potential therapeutic target. In clinical trials, PDE10A inhibitors, such as TP-10 and MP-10 (PF-2545920), have demonstrated safety and efficacy for cognitive and motor control improvement, particularly in brain disorders [15,16]. Despite this, the role of PDE10A in regulating NLRP3 inflammasome activation and its involvement in inflammasome-related diseases remains poorly understood.

In this study, we examined the effects of PDE10A inhibition on NLRP3 inflammasome activation and pyroptosis in vitro, using human THP-1 macrophages and mouse-derived macrophages, and in vivo, using sepsis [17,18] and sciatic nerve injury mouse models [19,20]. We utilized two PDE10A inhibitors, MP-10 and TP-10, which have been systemically administered in clinical trials for neuropsychiatric conditions like schizophrenia [21], as well as in preclinical models of heart failure [22,23]. Our results show that PDE10A inhibition effectively reduces inflammasome activation, protects against pyroptosis, and improves functional recovery in both sepsis and traumatic nerve injury models. These findings support the potential for targeting PDE10A in the treatment of inflammasome-driven diseases and suggest that PDE10A inhibitors may offer therapeutic benefits in managing inflammation and promoting recovery in various pathological conditions.

## 2. Results

To investigate the regulation and function of phosphodiesterases (PDEs) in immune cells, we previously screened PDE genes in peritoneal and alveolar macrophages. Our results identified PDE4B and PDE10A as the most significantly induced by lipopolysaccharide (LPS) [24]. PDE4 inhibitor Roflumilast is FDA-approved for treating severe chronic obstructive pulmonary disease (COPD) [25], but the specific role of PDE10A in NLRP3 inflammasome activation remains unclear. To explore this, we used two different PDE10A inhibitors, MP-10 and TP-10, in three types of macrophages: human THP-1 differentiated macrophages, mouse peritoneal macrophages, and mouse bone marrow-derived macrophages (BMDMs).

NLRP3 inflammasome activation involves two key steps: priming (e.g., NF-κB-mediated expression of NLRP3) and subsequent assembly and activation (e.g., by nigericin or ATP to activate caspase cleavage) [6]. The inflammasome then activates caspase-1, which processes pro-inflammatory cytokines like IL-1β and cleaves gasdermin D (GSDMD). This cleavage results in the formation of pores in the cell membrane, causing cell swelling and rupture, a process called pyroptosis [26,27].

To assess the role of PDE10A in inflammasome-mediated pyroptosis, we treated human THP-1 differentiated macrophages with LPS, with or without MP-10, followed by nigericin. Cell death was quantified using real-time nucleic acid staining (SYTOX™) [18]. Treatment with the PDE10A inhibitor MP-10 significantly reduced LPS/nigericin-induced cell death in a dose-dependent manner (Figure 1A), suggesting that PDE10A inhibition protects macrophages from pyroptosis.

Pyroptosis is a form of programmed cell death that amplifies inflammation, driven by inflammasome activation. This triggers caspase-1, which processes pro-inflammatory cytokines like IL-1β, intensifying both the inflammatory response and pyroptotic cell death. BMDMs were stimulated with LPS (100 ng/mL) for 3 h in the presence of MP-10 (5 μM), TP-10 (5 μM), or vehicle (DMSO), followed by an ATP (2 mM) challenge for 1 h. IL-1β levels in the culture medium were quantified by ELISA, showing that both MP-10 and TP-10 treatment reduced IL-1β secretion (Figure 1B).

During NLRP3 inflammasome activation, the adaptor protein apoptosis-associated speck-like protein containing a CARD (ASC) is recruited by NLRP3 and forms large multimeric complexes known as ASC specks, which are a necessary step for caspase recruitment and cleavage of inflammatory cytokines [18,28]. To examine the contribution of PDE10A to inflammasome activation, THP-1-derived macrophages were engineered to express an ASC-GFP fusion protein. Following stimulation with LPS and subsequent exposure to nigericin, cells were treated with the PDE10A inhibitor TP-10. This treatment led to a noticeable reduction in ASC speck formation, as observed through diminished GFP fluorescence, suggesting that PDE10A activity is required for optimal inflammasome assembly (Figure 1C).

To evaluate the effect of PDE10A inhibition on inflammasome activation, we measured the levels of cleaved caspase-1, cleaved gasdermin D (GSDMD), and mature IL-1β in macrophages treated with TP-10. GSDMD, a pore-forming protein, serves as the final effector in the activation of the inflammasome pathway, downstream of caspases-1, -4, and -11 [29]. NLRP3 inflammasome activation triggers caspase-1 cleavage, resulting in the activation of caspases that cleave both GSDMD and IL-1β. This cleavage generates an N-terminal fragment of GSDMD (GSDMD-NT) that forms transmembrane pores, facilitating the release of mature IL-1β and driving pyroptosis [30]. Treatment with TP-10 or MP-10 significantly reduced the levels of cleaved caspase-1, cleaved GSDMD, and mature IL-1β (Figure 1D and Appendix A), supporting the role of PDE10A in regulating inflammasome activation and pyroptosis.

Activated macrophages exhibit a significant increase in glycolytic activity, suggesting that PDE10A inhibition could potentially influence the expression of genes involved in glucose metabolism. Specifically, glucose transporter 1 (GLUT1) and hexokinase 1 (HK1) are key regulators of glucose metabolism in LPS-treated macrophages. After 3 h of LPS treatment, IL-6 mRNA expression was elevated by 20,000-fold compared to control. However, no significant changes were observed in the gene expression of GLUT1 or HK1. Treatment with the PDE10A inhibitor MP-10 significantly reduced LPS-induced IL-6 expression (Figure 2A) but had no effect on the expression levels of GLUT1 or HK1 (Figure 2B,C). These findings suggest that PDE10A does not play a role in modulating LPS-induced glucose metabolism at the transcriptional level.

The NLRP3 inflammasome can be activated by both sterile and non-sterile stimuli due to its capacity to detect a wide range of danger signals [31]. Non-sterile stimuli, such as pathogens or pathogen-associated molecular patterns (PAMPs) [32], and sterile stimuli, including tissue damage or metabolic stress, activate the inflammasome through shared downstream signaling pathways [33,34]. To investigate the role of PDE10A in inflammasome activation in vivo, we employed a clinically relevant acute sepsis model, using LPS followed by ATP as a second signal. Mice were treated with MP-10 after LPS injection, and three hours later, ATP was administered to induce NLRP3 inflammasome activation. MP-10 treatment significantly reduced the severity of sepsis, as measured by the murine sepsis scoring (MSS) system (Figure 3A), a reliable predictor of disease progression and mortality [35,36]. Plasma and peritoneal fluid were harvested 6 h after ATP injection, revealing significantly lower IL-1β levels in both fluids in MP-10-treated mice compared to DMSO-treated controls (Figure 3B,C). Thrombocytopenia, a common complication in sepsis associated with poor outcomes, was also mitigated by MP-10, which was shown to significantly increase total platelet counts. Additionally, MP-10 treatment reduced fibrin formation in the spleen, a hallmark of the inflammatory response in sepsis (Figure 3D). These findings collectively demonstrate that PDE10A regulates inflammasome activation and the inflammatory response in a mouse model of non-sterile sepsis.

Traumatic nerve injury causes mechanical damage followed by an inflammatory response, leading to axonal degeneration and demyelination [19,20,37]. Numerous studies have shown that the activation of the NLRP3 inflammasome is a major inflammatory response following spinal cord injury [38,39] and traumatic brain injury [40,41]. To investigate the role of PDE10A in this process, we employed a sterile sciatic nerve Injury (SNI) model to induce NLRP3 inflammasome activation. We hypothesize that inhibiting the NLRP3 inflammasome with a PDE10A inhibitor will reduce inflammation and improve motor function following SNI. Immunohistochemistry (IHC) performed on the sciatic nerve three days post-injury revealed a significant upregulation of PDE10A expression in response to nerve injury (Figure 4A). To assess the impact of nerve injury on inflammasome-related gene transcription, we collected sciatic nerve tissue three days after injury and quantified mRNA levels of inflammasome-associated genes, including NLRP2, ASC, NIAP5, NIAP6, and NLRP3. Among these, ASC and NLRP3 showed the most substantial increases in expression in injured nerves when compared to uninjured controls (Figure 4B). These findings suggest that both PDE10A and inflammasome signaling are involved in the pathophysiology of traumatic nerve injury.

To examine the effect of PDE10A inhibition on motor function recovery following nerve injury, we administered MP-10 or a vehicle control subcutaneously, starting three hours after the injury and continuing once daily for seven days. Motor recovery was assessed using the CatWalk gait analysis, and the sciatic functional index (SFI) was calculated based on walking footprint morphology on days 1, 7, and 14 post-injury (normal SFI range: 0–10; impaired range: 80–100). These results suggest that MP-10 enhances recovery following injury. The lack of difference between groups on day 1 and day 14 is expected—early after injury (day 1), recovery has not begun, and by day 14, both groups appear to be recovered. In contrast, the significant improvement in the MP-10 group on day 7, along with a significant time-by-treatment interaction, indicates that MP-10 accelerates recovery during the intermediate phase. Rather than reflecting a lack of efficacy, the absence of a difference on day 14 supports the interpretation that MP-10-treated mice recovered earlier than controls. (Figure 5A).

Nerve injury also triggers the transcriptional regulation of muscle atrophy-related genes, such as Myogenin [42] and Gadd45a [43,44], which are critical regulators of muscle-related diseases and neuromuscular impairments. We observed that MP-10 treatment significantly reduced the expression of Myogenin and Gadd45a in the gastrocnemius muscle three days after crush injury compared to the vehicle-treated group (Figure 5B-C).

To investigate the effects of PDE10A inhibition on inflammasome activation, we analyzed cleaved IL-1β levels in the sciatic nerve by IHC three days after crush injury. Nerve injury induced a marked increase in IL-1β cleavage in the sciatic nerve (Figure 5D). Importantly, MP-10 treatment significantly reduced IL-1β cleavage compared to the vehicle, indicating that PDE10A inhibition attenuates inflammasome activation (Figure 5D). These results collectively suggest that PDE10A inhibition modulates inflammasome activation and improves functional recovery following traumatic nerve injury.

## 3. Discussion

This study provides compelling evidence that PDE10A plays a pivotal role in regulating inflammasome activation and pyroptosis in macrophages. Using both in vitro and in vivo models, we demonstrated that the inhibition of PDE10A reduces inflammasome activation, mitigates inflammation, and improves functional recovery in conditions such as traumatic nerve injury and sepsis. Our findings suggest that PDE10A inhibitors, such as MP-10, may have therapeutic potential for treating diseases driven by excessive inflammation and inflammasome activation.

One key observation was that PDE10A inhibition significantly reduced cell death in macrophages activated by the NLRP3 inflammasome. In a dose-dependent manner, the PDE10A inhibitor MP-10 protected human THP-1 macrophages from pyroptosis induced by LPS and nigericin. This suggests that PDE10A contributes to regulating inflammasome-induced cell death, and its inhibition may offer a protective effect in inflammatory conditions. The inflammasome pathway involves key events, including NLRP3 priming, ASC speck formation, and the activation of caspase-1 and gasdermin D (GSDMD). Activation of the NLRP3 inflammasome cleaves pro-caspase-1, leading to IL-1β processing and GSDMD cleavage. The resulting GSDMD fragment (GSDMD-NT) forms membrane pores, triggering cell rupture and the release of inflammatory mediators. Several studies have indicated that cAMP regulates NLRP3 inflammasome activation by inhibiting its assembly through multiple mechanisms, including cAMP-PKA inhibition of NF-ĸB activity [45], direct cAMP binding to NLRP3 [46], and inhibition of caspase-1 activation [47]. Our results showed that MP-10 or TP-10 treatment significantly reduced cleaved caspase-1, cleaved GSDMD, and mature IL-1β levels, indicating that PDE10A is involved in regulating inflammasome activation. Furthermore, MP-10 inhibited ASC speck formation, a hallmark of inflammasome activation, suggesting that PDE10A modulates the assembly and activation of the NLRP3 inflammasome, but not downstream caspase-1 or GSDMD activity. These data underscore PDE10A’s role in regulating key steps of inflammasome activation and its potential to influence the severity of inflammasome-mediated inflammation.

Excessive inflammasome activation and pyroptosis can exacerbate inflammation, contributing to the progression of inflammatory diseases [30]. In a clinically relevant acute sepsis model, PDE10A inhibition reduced inflammation and improved sepsis outcomes. Treatment with MP-10 led to a lower murine sepsis score (MSS), reduced plasma IL-1β levels, alleviated thrombocytopenia, and decreased fibrin deposition in the spleen. These findings indicated that PDE10A inhibition modulates inflammasome activation and improves inflammatory markers and organ damage in sepsis. Given that inflammasome overactivation is central to sepsis pathophysiology [48], PDE10A represents a promising therapeutic target for diseases driven by inflammation, like sepsis.

Additionally, PDE10A inhibition improved motor function recovery following traumatic nerve injury. In a sciatic nerve crush injury model, MP-10 treatment significantly enhanced motor recovery, as measured by the sciatic functional index (SFI), compared to vehicle-treated mice. This improvement was further supported by reduced expression of muscle atrophy-related genes, including Myogenin and Gadd45a, in the gastrocnemius muscle. By inhibiting PDE10A, MP-10 not only attenuated inflammasome activation but also mitigated some of the downstream effects of nerve injury.

PDE10A expression has been associated with various injuries, including traumatic brain injury (TBI) [49,50,51,52] and spinal cord injury [53]. Consistent with our findings, PDE10A was highly induced after sciatic nerve crush injury. Emerging evidence also suggests that PDE10A inhibition can provide neuroprotection in TBI by reducing neuroinflammation and apoptosis via the cAMP/PKA pathway and downregulating NLRP3 inflammasome expression [50]. In our study, the reduced IL-1β cleavage in the sciatic nerve following MP-10 treatment further supported the idea that PDE10A inhibition modulates inflammasome activation in nerve injury, contributing to improved functional recovery and reduced muscle atrophy.

These findings highlighted PDE10A as a critical regulator of inflammasome activation and neuroinflammation, with its inhibition offering therapeutic benefits in both immune-related disorders and recovery from traumatic nerve injury. While our study provides strong evidence that PDE10A inhibition modulates inflammasome activation and inflammation, the precise molecular mechanisms remain unclear. Future research should confirm the role of PDE10A using genetic knockout models, investigate the downstream signaling pathways affected by PDE10A, and explore its function in other immune cell types. Additionally, although our in vivo models suggest therapeutic potential for treating conditions such as sepsis and traumatic nerve injury, additional long-term studies are needed to assess potential chronic side effects, particularly given the widespread expression of PDE10A across various cell types.

In conclusion, our study identified PDE10A as a key regulator of inflammasome activation and neuroinflammation. Inhibition of PDE10A reduced inflammasome-induced cytokine release, attenuated pyroptosis, and improved recovery following nerve injury and sepsis. These findings suggest that PDE10A represents a promising therapeutic target for treating inflammatory diseases and neuromuscular injuries driven by inflammasome activation. Although several PDE10A inhibitors have been tested for safety in humans, mainly for psychiatric disorders [54], further research is required before advancing to clinical trials for other conditions such as sepsis and trauma. Specifically, comprehensive toxicity evaluations that address long-term use and potential immunosuppressive effects, along with multi-dose efficacy studies in animal models, are critical.

## 4. Materials and Methods

### 4.1. Animal

Experiments were conducted using female and male C57BL/6J mice (Jackson Laboratory, Bar Harbor, ME, USA) between the ages of 10 and 14 weeks. The mice were housed in a specific pathogen-free environment with a 12 h light/dark cycle in a temperature-controlled room. They had unrestricted access to standard mouse chow and water. All animals were included in the study with no exclusions. Researchers conducting marker assessments were blinded to the experimental groups.

### 4.2. In Vivo Sepsis Model, MP-10 Treaments, and Tissue Harvesting

Sepsis was induced in C57BL/6J mice by intraperitoneal (IP) injection of LPS (10 mg/kg) (Sigma-Aldrich, St. Louis, MO, USA) for 3 h, followed by an IP injection of 100 mM ATP (100 µL, pH 7.4) (Sigma-Aldrich, A7699), as previously described [17,18]. Mice were randomly assigned to experimental groups. MP-10 was dissolved in a vehicle containing 10% DMSO and 40% hydroxypropyl-β-cyclodextrin, based on previous studies [22]. Mice were given a subcutaneous (S.C.) injection of either vehicle or MP-10 (3 mg/kg) 30 min after LPS administration. Blood and peritoneal fluid samples were collected 30 min after ATP injection. Mice were administered an intraperitoneal (IP) injection of ketamine and xylazine for deep anesthesia. Whole blood was drawn into tubes containing 10 µL of 500 mM EDTA, then subjected to centrifugation at 1000× *g* for 10 min at 4 °C. Plasma was collected and aliquoted for cytokine assays. Peritoneal fluid was obtained by irrigating the peritoneal cavity with 5 mL of ice-cold PBS containing 1 mM EDTA. The cell suspension was centrifuged at 500 g for 5 min at 4 °C, and the supernatant was collected and aliquoted for cytokine assays. Spleen tissues were harvested and fixed in 4% Paraformaldehyde (PFA) (VWR International, Radnor, PA, USA) overnight at 4 °C, followed by embedding in paraffin.

### 4.3. Sciatic Nerve Crush and Denervation Injuries, MP-10 Treatments, and Tissue Harvesting

Sciatic nerve crush injury was performed on female C57BL/6J mice as previously described [19,20]. Briefly, Buprenorphine ER (0.5–1 mg/kg) was administered subcutaneously 1 h prior to surgery for effective postoperative pain relief lasting 48–72 h. After inducing anesthesia with intraperitoneal (IP) injections of ketamine (60 mg/kg) and xylazine (4 mg/kg), the mice were prepared aseptically. A lateral skin incision was made along the femur, and the sciatic nerve was bluntly exposed. The nerve was crushed proximal to the tibial and peroneal branches using smooth forceps with a metal calibration ring to standardize the pressure for 30 s. The wound was then closed, and the mice were monitored every 12 h for the next 3 days. For MP-10 treatment, mice were randomized to receive daily subcutaneous injections of either MP-10 (10 mg/kg) or vehicle (10% DMSO in 40% β-cyclodextrin *v*/*v*) starting 1 h after injury and continuing until day 7. On the day of sacrifice, the mice were given an intraperitoneal (IP) injection of ketamine and xylazine for deep anesthesia. Blood samples, sciatic nerves, and muscle tissue were subsequently collected. Gastrocnemius–soleus (GAS) muscles were harvested for gene expression analysis.

### 4.4. Sciatic Functional Index (SFI)

The effects of MP-10 treatment were evaluated using the Sciatic Functional Index (SFI), a noninvasive method for assessing in vivo functional recovery following sciatic nerve injury [19,20]. The SFI is scored on a scale from 0 (normal function) to 100 (complete loss of function). Briefly, mice were trained to walk freely along a 77 cm by 7 cm corridor lined with white paper, and individual paw prints were obtained by painting each hind foot. Paw prints were measured by two blinded evaluators for toe spread (distance from the 1st to the 5th toe) and paw length (distance from the 3rd toe to the bottom of the print). Three prints from both the experimental (injured) and normal (uninjured) sides were measured, and the SFI was calculated for each animal by averaging these measurements and applying the following formula [55]: SFI = 118.9 ((ETS-NTS)/NTS)) − 5 1.2 ((EPL-NPL)/NPL)) − 7.5. Where E refers to the experimental (injured: crushed or denervated) paw, N refers to the normal (healthy: uninjured or control) paw, TS is the toe spread, and PL is paw length. To ensure rigor and reproducibility, both testers and scorers were blinded to all treatment conditions.

### 4.5. Bone Marrow Progenitor Cell Isolation and Bone Marrow-Derived Macrophage (BMDM) Differentiation

BMDM preparation was performed, as previously described [17,18]. L929 conditioned media, which contains the macrophage growth factor M-CSF, was prepared by culturing L929 cells (ATCC) in complete DMEM (Thermo Fisher Scientific, MA, USA) supplemented with 10% FBS, and 1% penicillin and streptomycin for 10 days at 37 °C, 5% CO_2_. The L929 conditioned media, was collected, filtered (Corning, New York City, NY, USA), and stored at −80 °C until required. For isolation of BMDMs, tibias and femurs were removed from both C57BL/6J male and female mice. Bone marrow was harvested by centrifuging at 500× *g* for 2 min at 4 °C, then resuspended in a complete DMEM medium and passed through a 70-μm cell strainer (VWR International, Radnor, PA, USA). The bone marrow progenitor cells were cultured in 100 mm dishes for 6–7 days in a medium consisting of 70% complete DMEM and 30% L929-conditioned medium. Fresh medium (5 mL) was added on day 3. BMDMs were collected by scraping in cold PBS containing EDTA (1 mM). After centrifugation, BMDMs were seeded into 12-well plates at a density of 1.6 × 10^5^ cells/well in DMEM and incubated overnight before use.

### 4.6. THP-1 Macrophage Differentiation

Human THP-1 monocytes were differentiated into macrophages by incubating them with 100 nM PMA (Sigma-Aldrich, St. Louis, MO, USA) in a complete RPMI medium for 24 h at a density of 1.6 × 10^5^ cells per well in 12-well plates. After two washes with 1x PBS, the cells were cultured in a complete RPMI medium without PMA for an additional 24 h prior to the experiment.

### 4.7. ASC-GFP-Overexpressed THP-1 Macrophage

A lentivirus expressing the ASC-GFP fusion protein was generated by co-transfecting pLEX-MCS-ASC-GFP (Addgene, Cambridge, MA, USA), psPAX2, and pMD.2G into HEK293T cells using Fugene 6 (Promega, Madison, WI, USA). After 24 h, the culture medium was replaced with fresh medium containing 5% FBS. The virus-containing supernatant was harvested 48 h later and then stored at −80 °C. To infect THP-1 monocytes (1.6 × 10^5^ cells per well in 12-well plates), 400 µL of virus-containing medium was mixed with polybrene (4 µg/mL) and centrifuged at 2500 rpm for 90 min at 20 °C. Following infection, a fresh complete RPMI medium with 100 nM PMA was added, and the cells were incubated for 24 h. After two washes with 1x PBS, the cells were incubated for an additional 24 h in a complete RPMI medium without PMA before proceeding with the experiment.

### 4.8. Peritoneal Macrophage Isolation

Peritoneal macrophage isolation was performed, as previously described [24]. Bio-Gel preparation: Wash Bio-Gel P-100 gel three times with PBS. Pellet Bio-Gel by centrifugation for 5 min at 400× *g*, then resuspend in PBS to yield 2% (*w*/*v*) Bio-Gel. Autoclave for 20 min before use. C57BL/6J mice between 10 and 12 weeks of age were used. Mice received an intraperitoneal injection (IP) of 1mL 2% fine polyacrylamide beads. On day four after the injection, the animals were euthanized by CO2, and macrophages were harvested by washing their peritoneal cavity with 5 mL ice-cold PBS supplemented with 1mM EDTA. The cell suspension was centrifuged at 500× *g* for 5 min at 4 °C, and the supernatant was discarded. The cell pellet was washed one time in PBS and the supernatant was removed through centrifugation. The cell pellet was suspended in RPMI 1640 medium supplemented with 10% FBS, 1% streptomycin/penicillin, 0.1mM 2-Mercaptoethanol, and 1mM sodium pyruvate. Macrophages were cultured at 1.6 × 10^5^ cells/well on 12-well plates. After incubating for 2 h, cells were washed twice with PBS, and the media were replenished. Cells were used for experiments after 24 h of culture.

### 4.9. Inflammasome Stimulation

To stimulate the NLRP3 inflammasome, cells were first treated with LPS (100 ng/mL) for 3 h to induce priming. Afterward, nigericin (2 µM for BMDMs and peritoneal macrophages, 6 µM for THP-1-differentiated macrophages; Sigma-Aldrich, N7143-5MG) or ATP (2 mM) was added for 30 min to activate the inflammasome. MP-10 (3–5 µM; MedChemExpress, Monmouth Junction, NJ, USA) or TP-10 (3–5 µM; MedChemExpress, Monmouth Junction, NJ, USA) was co-administered with LPS during priming.

### 4.10. Cell Death by SYTOX^TM^ Green

THP-1 macrophages or BMDMs were seeded in 96-well plates (2 × 10^4^ cells/well) one day before the experiments. Cells were washed twice and incubated with LPS (100 ng/mL) in XF based medium (Agilent, Santa Clara, CA, USA) supplemented with 4.5 g/L glucose, 2 mM glutamine, 1 mM sodium pyruvate, and 1 mM HEPES buffer at final pH of 7.4 for 3 h. After 3 h of LPS stimulation, SYTOX Green (final concentration 1µM) (Thermo Fisher Scientific, Waltham, MA, USA) was added together with nigericin (2 µM for BMDMs; 6 µM for THP-1 macrophages) and fluorescence signals (Excitation wavelength: 485 nm, Emission wavelength: 550 nm) were analyzed using FLUOstar OPTIMA plate reader (BMG Labtech, Baden-Württemberg, Germany) at 36 °C for 120 min. The percentage of cell death was calculated by normalizing fluorescence signals from cells treated with Triton X-100 (0.1%).

### 4.11. Cytokine Assays

IL-1β levels in plasma and peritoneal fluid from mouse studies were quantified using ELISA kits following the manufacturer’s guidelines. For in vitro experiments, culture supernatants were collected immediately after treatment, then centrifuged at 16,000× *g* for 5 min to remove debris. The cleared samples were stored at −20 °C. Mouse IL-1β levels (BioLegend, San Diego, CA, USA) in the supernatants were then measured using an ELISA assay.

### 4.12. RNA Extraction and Real-Time PCR

RNA was isolated from tissues or cultured cells using the RNeasy kit (Qiagen, Hilden, Germany) following the manufacturer’s protocol. Complementary DNA (cDNA) was synthesized from 0.5 μg of RNA using the iScript™ cDNA Synthesis Kit (Bio-Rad, Hercules, CA, USA). For amplification, cDNA and 1 μM forward and reverse primers were combined with iQ™ SYBR^®^ Green Supermix (Bio-Rad, Hercules, CA, USA). Fluorescence data were collected and analyzed using a CFX Connect real-time PCR system (Bio-Rad). Gene expression levels were normalized to β-actin using the delta–delta cycle threshold method. Specific transcript amplification was verified through melting curve analysis performed at the end of each PCR run. The following primers were used: Mouse GLUT1 (Forward: GGTTGTGCCATACTCATGACC, Reverse: CAGATAGGACATCCAGGGTAGC); Mouse HK1 (Forward: GTGGACGGGACGCTCTAC, Reverse: TTCACTGTTTGGTGCATGATT); Mouse β-actin (Forward: TTCAACACCCCAGCCATGT, Reverse: GTAGATGGGCACAGTGTGGGT); Mouse NLRP3 (Forward: TTCCCAGACACTCATGTTGC, Reverse: AGAAGAGACCACGGCAGAAG); Mouse Myogenin (Forward: CCTTGCTCAGCTCCCTCA, Reverse: TGGGAGTTGCATTCACTGG); Mouse Gadd45a (Forward: AGAGCAGAAGACCGAAGGA, Reverse: CGTAATGGTGCGCTGACTC); Mouse ASC (Forward: CTGCTCAGAGTACAGCCAGAAC, Reverse: CTGTCCTTCAGTCAGCACACTG); Mouse NLRP2 (Forward: TCTCATGTGCCTTCTACATCAGC, Reverse: CACAAAGGTCACGGATCAGAG); Mouse NAIP5 (Forward: TGCCAAACCTACAAGAGCTGA, Reverse: CAAGCGTTTAGACTGGGGATG); Mouse NAIP6 (Forward: TACAGGGAGTTTACAAGACCCC, Reverse: AGTGGCCTGGAGAGACTCAG).

### 4.13. Western Blot

Proteins were separated by SDS-PAGE using 10% acrylamide gels and transferred to nitrocellulose membranes. The membranes were blocked with 5% nonfat dry milk in Tween-TBS, then incubated overnight with the following primary antibodies: Caspase-1 (AdipoGen, Liestal, Switzerland) at 1:1000, GSDMD (Abcam, Cambridge, UK) at 1:1000, IL-1β (GeneTex, Irvine, CA, USA) at 1:2000, and β-actin (Cell Signaling Technology, Danvers, MA, USA) at 1:1000. After washing, the membranes were incubated with HRP-conjugated secondary antibodies: Anti-mouse IgG (Cell Signaling Technology, 7076), Anti-rabbit IgG (Cell Signaling Technology, Danvers, MA, USA), or Anti-goat IgG (Jackson ImmunoResearch, West Grove, PA, USA). Protein bands were visualized using enhanced chemiluminescence, exposed to radiographic film, and quantified by densitometry, with the results normalized to β-actin levels from the same gel. All of the full and uncropped Western blots were listed in Appendix A.

### 4.14. Immunofluorescence Analysis

Frozen and OCT-embedded sciatic nerves and Tibialis Anterior (TA) muscles from each group were sequentially cross-sectioned at 10 μm and stored at −80° C. Tissue sections were incubated with 0.2% Triton X-100 for 10 min and blocked in 10% normal goat serum (Jackson ImmunoResearch, West Grove, PA, USA) > for 30 min at room temperature. After overnight incubation with primary antibody (either 1:1000 PDE10A (Santa Cruz, Dallas, TX, USA), or 1:1000 cleaved IL-1β (Thermo Fisher Scientific, Waltham, MA, USA) at 4 °C, the fluorescent-labeled secondary antibody was incubated for 1 h at room temperature. Spleen tissues were drop-fixed in 4% PFA, embedded in paraffin, cut into 5 μm sections, and mounted onto slides. Sections were deparaffinized before use. Sections were washed 3 times in PBS followed by antigen retrieval for 20 min with steam using 1X Citrate buffer (Millipore, Burlington, VT, USA), PH = 6.0. Tissue sections were incubated with 10% normal goat serum (Vector Laboratories, Newark, NJ, USA) in PBS for 1 h at room temperature to block non-specific binding. Afterward, they were incubated overnight at 4 °C with Fibrin (Millipore, Burlington, MA, USA) diluted in 2% normal goat serum in PBS. After three washes with PBS, fluorescence-conjugated secondary antibodies (Molecular Probes, Eugene, OR, USA) 1:1000 were incubated for 1 h at room temperature and followed by three washes with PBS. All slides were mounted with DAPI Fluoromount-G (SouthernBiotech, Birmingham, AL, USA) to detect nuclei.

### 4.15. ASC Speck Formation

ASC-GFP-labeled THP-1 macrophages, differentiated in vitro, were plated onto glass coverslips in 12-well plates. The cells were exposed to 100 ng/mL LPS, with or without TP-10 (5 µM), for 3 h, then treated with 6 µM nigericin for 2 h. Cells were fixed in 4% PFA for 10 min. After three washes with 1× PBS, cells were mounted using fluoromount-G-DAPI. Immunofluorescence was analyzed by a confocal microscope. The GFP 488 nm fluorescent signals were acquired by confocal microscopy. The data points represent biological replicates from 3 separate experiments.

### 4.16. Statistics

Unless otherwise specified, in vitro experiments were performed in triplicate, with results from duplicate or triplicate wells averaged before statistical analysis. Data are presented as mean ± SEM. Statistical analysis was conducted using GraphPad Prism 10.0. For cell culture experiments with PDE10A inhibitors, data were analyzed using one-way ANOVA. In vivo data from MP-10 treatment in sepsis and sciatic nerve injury models were assessed using two-way ANOVA, followed by Bonferroni-corrected post hoc *t*-tests for multiple comparisons. *p* values are indicated as follows: * <0.05, ** <0.01, *** <0.001, **** <0.0001.

## 5. Conclusions

Our findings emphasize the role of PDE10A in inflammasome activation and pyroptosis in macrophages. Inhibition of PDE10A with MP-10 and TP-10 reduced cytokine release and cell death, suggesting potential therapeutic benefits for modulating inflammation and improving recovery in conditions like traumatic nerve injury and sepsis.

## Figures and Tables

**Figure 1 ijms-26-04498-f001:**
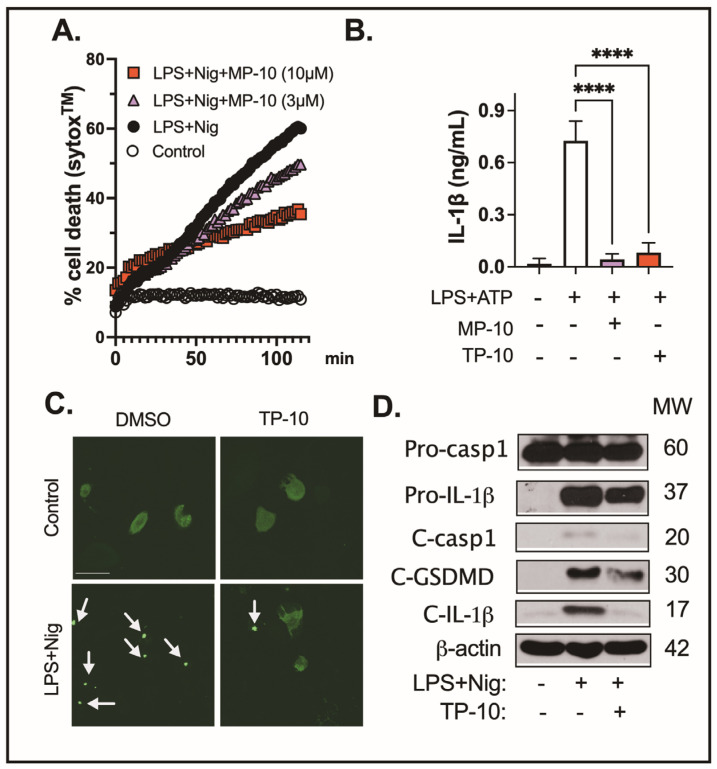
PDE10A inhibition reduces NLRP3 inflammasome activation in macrophages. (**A**) THP-1-derived macrophages were treated with LPS (100 ng/mL) and either MP-10 (3 or 10 μM) or vehicle (DMSO) for 3 h, followed by a 2 h stimulation with nigericin (6 μM). Real-time percentage cell death was determined by normalizing to maximal nuclear acid fluorescent intensity (0.1% Triton-X100 treated cells). (**B**) BMDMs were treated with LPS (100 ng/mL) for 3 h in the presence of MP-10 (5 μM), TP-10 (5 μM), or vehicle (DMSO), followed by a 1 h stimulation with ATP (2 mM). IL-1β levels in the culture medium were quantified by ELISA. (**C**) THP-1 ASC-GFP overexpressing macrophages were stimulated with LPS (100 ng/mL) and nigericin (Nig, 6 μM) with or without TP-10 (5 μM). ASC speck formation was measured by confocal microscopy. Arrowheads indicate ASC specks (green). Scale bar = 50 µm. (**D**) Peritoneal macrophages were treated with LPS (100 ng/mL) for 3 h in the presence or absence of TP-10 (5 μM), followed by stimulation with nigericin (2 μM). Pro- and cleaved caspase-1, IL-1β, and GSDMD were measured by Western blotting. Data were expressed as mean ± SEM. Statistics in (**A**,**B**) were performed using one-way ANOVA. **** *p* < 0.001.

**Figure 2 ijms-26-04498-f002:**
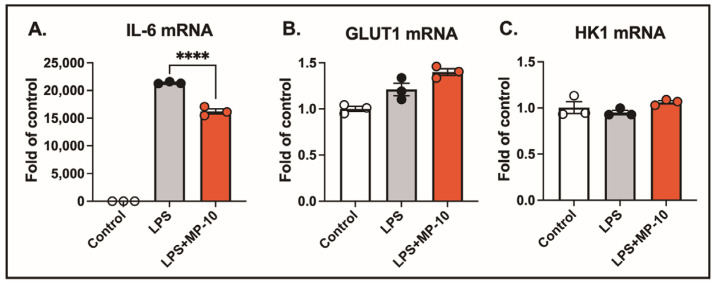
PDE10A does not regulate LPS-induced glucose metabolism at the transcriptional level. BMDMs were stimulated with LPS (100 ng/mL) for 3 h in the presence of MP-10 (5 μM) or vehicle (DMSO). mRNA expression in cell lysates was measured for (**A**) IL-6, (**B**) GLUT1, and (**C**) HK1. Data are expressed as mean ± SEM. Statistics in (**A**–**C**) were performed using one-way ANOVA. **** *p* < 0.001.

**Figure 3 ijms-26-04498-f003:**
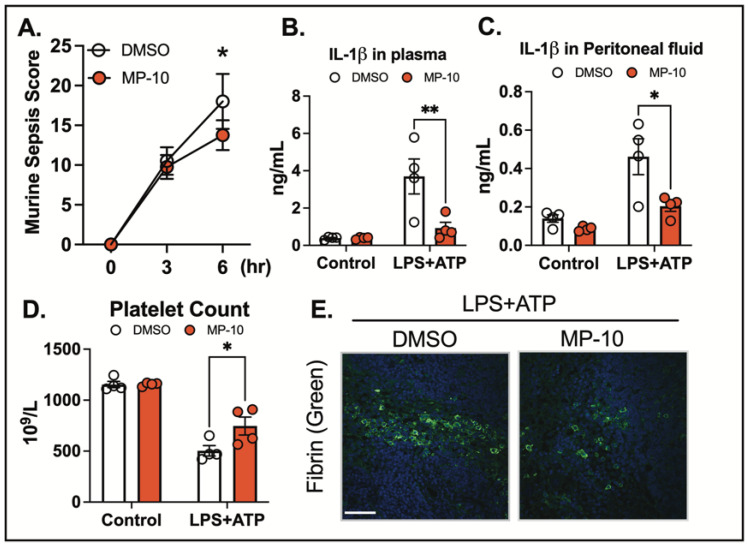
PDE10A inhibition reduces sepsis severity. C57BL/6J mice underwent acute LPS (10 mg/kg body weight, i.p.) treatment with or without MP-10 (3 mg/kg, s.c. injection) for 3 h, followed by ATP (100 mM in 100 µL, pH 7.4) i.p. (**A**) Murine sepsis score was measured at 3 and 6 h after ATP injection. (**B**) IL-1β in plasma, (**C**) IL-1β in peritoneal fluid, (**D**) Total platelet count in whole blood, (**E**) Fibrin (green), and DAPI (Blue) staining in the spleen. Scale bar = 60 µm. *n*= 4 for each group. Statistics in (**A**) were performed using one-way ANOVA. Statistics in (**B**–**D**) were performed using two-way ANOVA and Bonferroni’s post hoc test. * *p* < 0.05, ** *p* < 0.01.

**Figure 4 ijms-26-04498-f004:**
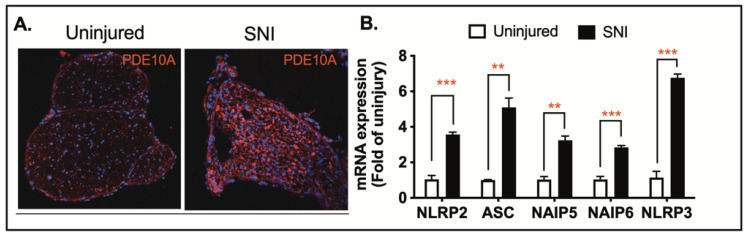
Sciatic nerve injury-induced PDE10A protein expression and inflammasome gene Expression. (**A**) C57BL/6N mice underwent sham surgery and peripheral nerve injury (PNI). Sciatic nerves were harvested on day 3 for PDE10A staining (PDE10A, red; DAPI, blue). Representative immunofluorescence images of 2 animals from each group are shown. (**B**) Gene expression of inflammasomes. *n* = 4 from each group. Data were expressed as mean ± SEM. Statistics in (**A**) were performed using one-way ANOVA. ** *p* < 0.01; *** *p* < 0.001.

**Figure 5 ijms-26-04498-f005:**
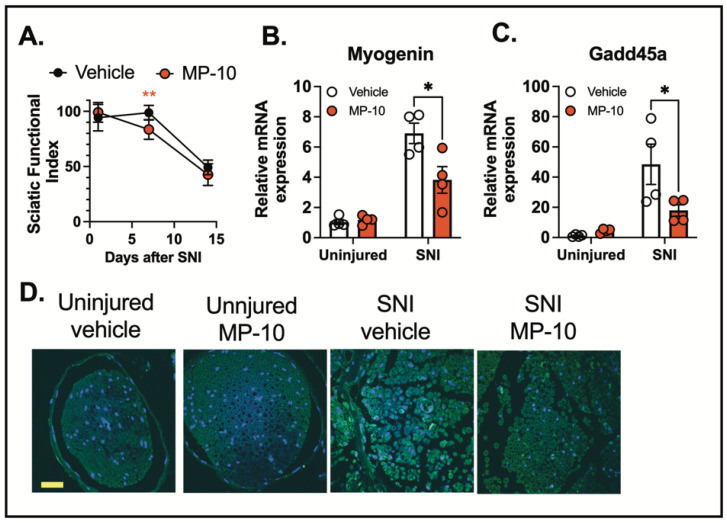
PDE10A inhibition reduces gene expression of muscle atrophy and inflammasome activation induced by sciatic nerve injury. (**A**) C57BL/6N mice underwent sciatic nerve crush injury and were treated with vehicle or MP-10 once daily (s.c. injection, 10 mg/kg). The sciatic functional index was measured by gait analysis on day 7 after injury *(n* = 8 for each group). (**B**) Myogenin mRNA expression, (*n* = 4 for each group), (**C**) Gadd45a mRNA expression in gastrocnemius muscle (*n* = 4 for each group), (**D**) Cleaved IL-1β staining (green) in the sciatic nerve. Representative immunofluorescence images of 4 animals from each group are shown. Scale bar = 200 µm. Data were expressed as mean ± SEM. Statistics in (**A**) were performed using one-way ANOVA. Statistics in (**B**,**C**) were performed using two-way ANOVA and Bonferroni’s post hoc test. * *p* < 0.05, ** *p* < 0.01.

## Data Availability

All relevant data supporting the findings of this study are available within the manuscript.

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
