# Peer review of "PDE10A Inhibition Reduces NLRP3 Activation and Pyroptosis in Sepsis and Nerve Injury"

_ijms, 2025, doi:10.3390/ijms26104498_

Round 1
Reviewer 1 Report
Comments and Suggestions for Authors
The manuscript explores if PDE10A inhibition will reduce NLRP3 activation and pyroptosis in specific inflammatory situations (sepsis and nerve injury). The authors attempted to answer the question with in vitro and in vivo experiments that are well thought out. The goal of the question and answer is to identify PDE10A as a therapeutic target. The article provides some evidence that support PDE10A inhibition may be beneficial in traumatic nerve disorder and in the treatment of sepsis, but the picture is far from complete and requires more downstream investigations.
The topic is relevant to the field and addresses a gap in knowledge about one of the PDEs. It aims at identifying an additional therapeutic target for the treatment of sepsis and traumatic nerve disorder. Identifying therapeutic targets is a relevant endeavor. The study reasonably suggests PDE10A inhibition as a desirable step that needs to be explored in a clinical setting.
Figure 3 seems to be truncated, and the legend is very short. It seems to be lost in the production or when the manuscript was put together. I can only see A and B when the text refers to other letters. Also mentioning what the authors did to ensure rigor and reproducibility in their experiments.
The conclusions are acceptable and are based on the experiments that answer the posed question. References are adequate.
Author Response
Reviewer1:
The manuscript explores if PDE10A inhibition will reduce NLRP3 activation and pyroptosis in specific inflammatory situations (sepsis and nerve injury). The authors attempted to answer the question with in vitro and in vivo experiments that are well thought out. The goal of the question and answer is to identify PDE10A as a therapeutic target. The article provides some evidence that support PDE10A inhibition may be beneficial in traumatic nerve disorder and in the treatment of sepsis, but the picture is far from complete and requires more downstream investigations.
The topic is relevant to the field and addresses a gap in knowledge about one of the PDEs. It aims at identifying an additional therapeutic target for the treatment of sepsis and traumatic nerve disorder. Identifying therapeutic targets is a relevant endeavor. The study reasonably suggests PDE10A inhibition as a desirable step that needs to be explored in a clinical setting.
Figure 3 seems to be truncated, and the legend is very short. It seems to be lost in the production or when the manuscript was put together. I can only see A and B when the text refers to other letters. Also mentioning what the authors did to ensure rigor and reproducibility in their experiments.
The conclusions are acceptable and are based on the experiments that answer the posed question. References are adequate.
Response:
- We appreciate the reviewer's comments. We notice that the figure 3 legend was correct, but Figure 3 and Figure 4 were misplaced. We have replaced them in the right order.
- To address rigor and reproducibility, we have included the following sentence" To ensure rigor and reproducibility, both testers and scorers were blinded to all treatment conditions." In lines 386-387.
Reviewer 2 Report
Comments and Suggestions for Authors
This study used multiple cell models and mouse models to demonstrate that PDE10A inhibitors can significantly reduce NLRP3 activation and the expression of IL-1β, caspase-1, and cleaved gasdermin D (GSDMD), and improve neurological function and sepsis indicators. The authors focus on the role of phosphodiesterases (PDEs) in NLRP3 inflammasome activation and pyroptosis. Therefore, MP-10 inhibitors were used for in vivo and in vitro experiments to demonstrate their potential to reduce inflammatory responses and improve functional recovery of nerve damage. The authors' core hypothesis is that PDE10A inhibition can regulate immune metabolism, reduce NLRP3 activation and inflammatory response, and thus improve the consequences of sepsis and peripheral nerve injury. Overall, this is a well-designed and well-documented study.
Comments:
- Although the authors used drugs MP-10 and TP-10 for inhibition, they did not use PDE10A knockout mice or siRNA knockdown technology to further verify its specificity and causal relationship. If combined with a gene editing model, it would provide more powerful evidence for the role of PDE10A in the NLRP3 pathway.
- The comparison group design was slightly insufficient. Most experiments only compared "LPS + ATP" vs. "LPS + ATP + MP-10", lacking the control of "ATP stimulation alone" or "inhibitor treatment alone", which easily overlooked the non-specific effects of the drug.
- Speculations about clinical applications are overly optimistic. Although MP-10 has been tested for safety in humans (for psychiatric disorders), the authors suggest that it is premature to move directly into clinical trials for sepsis. In the absence of complete toxicity analysis (such as long-term use, immunosuppressive side effects) and multi-dose efficacy curves, a more conservative discussion should be conducted.
- If MP-10 has entered the human trial stage, are there any long-term side effects to be evaluated? PDE10A is an important regulator of inflammation. PDE10A is widely expressed in multiple cell types. Is inhibition likely to have side effects?
- Figure 5 and results: The authors explored the effect of PDE10A inhibition on the recovery of motor function after peripheral nerve injury. The time point was only 7 days after injury, which is slightly insufficient. More different time points should be added for observation to better evaluate the effect of PDE10A inhibition on the recovery of motor function after peripheral nerve injury.

Round 2
Reviewer 2 Report
Comments and Suggestions for Authors
The authors have made appropriate revisions according to the reviewers' comments. I recommend the manuscript be published in Int. J. Mol. Sci.